# Family and School Relationship during COVID-19 Pandemic: A Systematic Review

**DOI:** 10.3390/ijerph182111710

**Published:** 2021-11-08

**Authors:** José Juan Carrión-Martínez, Cristina Pinel-Martínez, María Dolores Pérez-Esteban, Isabel María Román-Sánchez

**Affiliations:** 1Education Department, Universidad de Almería, 04120 Almería, Spain; jcarrion@ual.es (J.J.C.-M.); cpm467@ual.es (C.P.-M.); mpe242@ual.es (M.D.P.-E.); 2Economy and Enterprise Department, Universidad de Almería, 04120 Almería, Spain

**Keywords:** family, COVID-19, school, students, relation

## Abstract

Education systems worldwide have been affected by a sudden interruption in classroom learning because the coronavirus pandemic forced both the closure of all schools in March 2020 and the beginning of distance learning from home, thus compelling families, schools, and students to work together in a more coordinated fashion. The present systematic review was carried out following PRISMA guidelines. The main objective was to present critical information on the relationship between the family and the school in the face of the imposed distance learning scenario caused by COVID-19. A total of 25 articles dealing with the relationships established during the pandemic of any of the three agents involved (family, students, and school) were analysed. The results showed that the relationships between the three groups involved must be improved to some extent to meet the needs that have arisen as a result of distance learning. In conclusion, the educational scenario during the pandemic has been one of the most significant challenges experienced in the recent history of education.

## 1. Introduction

On 11 March 2020, the World Health Organization (WHO) [1] declared the new coronavirus outbreak (COVID-19) as a pandemic, which quickly ravaged the entire world from its epicentre in Wuhan, China, in December 2019 [2]. Given the virus’s rapid pace of expansion and the high rates of infection and mortality around the world [3,4], one of the measures taken by many nations and states was to place the civilian population under lockdown or quarantine measures, whose duration and characteristics were subject to the advance of the virus in each country [5]. Although these restrictive measures have been shown to have had positive effects against the spread of the virus [6,7,8], as was observed with previous diseases such as swine flu or MRSA [9,10], society was forced to enter a new reality that directly affected daily routines [11,12,13,14] and habits [15]. One of the most immediate government decisions was to close all educational institutions and opt for virtual or distance education from home [16,17].

This new educational landscape has led to each of the involved educational agents to perceive the same event in different ways. From the perspective of schools and teachers, it has been observed that despite some regions having kept face-to-face learning with fully open centres [18], educative centres from other countries have partially or totally closed, forcing students at all educational levels to turn their homes into learning centres in a very short period, thus disrupting their educational processes [15,19]. Schools from many regions were forced to move from purely face-to-face learning to blended learning or, in numerous cases, to purely virtual and distance learning [20]. Thus, the teaching–learning process radically changed, and both educational institutions and teachers had to immediately react, turning educational digital tools into educational elements par excellence throughout this process [21]. Accordingly, information and communication technologies (ICT) and learning and knowledge technologies (TAC) were converted from complementary tools, sometimes infrequently used by teachers in their classes, to a main and binding element of the teaching–learning process [21].

The closure of educational institutions over long periods of time has always been of interest to researchers and international organizations that are concerned with armed conflicts, strikes, or natural disasters and their subsequent consequences on educative centres. In such cases, it has often been observed that the acquisition of basic skills has been diminished, especially in students who come from disadvantaged backgrounds [22,23].

The coronavirus pandemic has engendered these kinds of situations, as the closure of educational institutions in several regions and countries and blended-learning imposition in many others have been generalized throughout the world. This situation has led to the realization of one of the most extensive global educational experiments in recent history, as various platforms, radio, and television channels became educational sources that could be accessed from within homes to allow students to keep learning [16,20,24]

In line with this observation and focusing on the perspective of the family environment, with schools closed, families were forced to assume a new role in the education of their children and have gone from being one of the educational agents [25,26] who (in collaboration with the school, looked after the interests and success of students) to assuming the roles of teachers and learning facilitators [27,28].

The many case studies concerning these obligatory relationships that quickly had to be assumed by schools, families, and students have revealed the many needs and difficulties that had to be faced, including a disparity of economic resources [29], lack of internet accessibility [30], lack of digital skills [31], and the inability of families to provide curricular help [32].

The essential role of families in the teaching–learning processes was already revealed by a great variety of studies carried out before pandemic [33], which showed that families are undoubtedly one of the main gears that guaranteed the success or failure of educational systems, as well as the development of significant educational activities [34].

Schools were forced to establish a double aspect of relationships with families [35,36] (who assumed the role of proxy teachers [37]) and students to continue developing the teaching–learning processes. Similarly, families had to internally strengthen relationships with their children for optimal educational processes.

Beyond the more institutional and family perspective, it is also important to observe how the change from face-to-face education to a distanced and semi-presential one has been experienced by students, the main protagonists in all teaching–learning processes.

Students have experienced the entire transition period from more traditional learning, having to move from their homes to educational institutions to a virtual or blended scenario in which the contents reached their homes without having to travel. This situation has generated situations of stress, anxiety, and uncertainty among students, not knowing when they would return to a period of “normality” to which they were accustomed [38].

Likewise, students’ motivation to study was also greatly affected by both the fact that parents acquired the role of teachers and the necessity to learn quickly, without hesitation, all the technological skills required to be able to access a remote education [39]. Considering that different virtual platforms have been used to develop each subject content and lessons, being able of controlling each one of them has meant an added effort for the students [40].

One of the most considerable challenges traditionally tackled by schools is the commitment to forging stronger bonds between the school and the families and between the students and families [35] by opting for more significant family presence and involvement [41]. Given that the results have not always been achieved as intended, many teachers have called for more robust connections and greater involvement from all educational community members [42]. In studies conducted before the pandemic, parental involvement in education was witnessed to be essential in children’s school performance [43,44,45]. Other studies have revealed that many adults have a great educational deficit concerning new technologies [45,46,47].

Nevertheless, the use of new technologies and the development of a virtual education, in which the relationships between educational institutions, families, and students must be very present, has become a new educational paradigm that is already far removed from the circumstantial situation originated by the pandemic. As a matter of fact, it has been currently implemented as a new educational model that brings to light the real resilient potential of the different educational systems, as well as its capacities to explore novel approaches and models that allow for the satisfaction of present needs [48].

In this context, we have asked ourselves questions such as (1) how supportive have the relationships between the family and the school and the school and the students been during the pandemic? (2) how have parents been involved in the task of educating their children during the pandemic? and (3) have these three groups (family, school, and students) supported each other sufficiently? These questions were linked to the more general objective of presenting critical information about the relationship established between the family and the school in the face of an imposed distance education scenario due to COVID-19. We break this objective down into the following specific objectives: (1) to analyse what relationships have been established between the family and the school, (2) to determine how the school has supported its students, and (3) to examine what relationships have been constituted between parents and their children in a home learning situation.

## 2. Materials and Methods

The present systematic review followed the Preferred Reporting Items for Systematic Reviews and Meta-Analyses (PRISMA) [49]. Our primary purpose was to select studies related to the response of families and schools to non-presential teaching scenarios resulting from COVID-19 measures.

### 2.1. Search Strategies

The authors of the present work carried out a literature review in several phases. First, a review of the generic literature was carried out in the main scientific research databases and specialized journals on health and educational issues, both national and international. In order to cover the largest number of studies related to the objectives set above, the Web Science (WoS) Scopus, Dialnet Plus, Centers for Disease Control and Prevention (CDC), The New England Journal of Medicine (NEJM), Science Direct, and ERIC databases were selected. During this phase, the descriptors used were, in Spanish, “COVID-19”; “familias”, “familiar”, “escuelas”, “educación”, “educativo”, “padres”, “progenitores”, “niño”, “niños”, and “adolescentes”. In English, the descriptors used were “COVID-19”; “Family”, “home”, “house school”, “education”, “educative”, “parent”, “school”, “child”, “children”, and “adolescent”. Where permitted, the Boolean operators “AND” and “OR” and apocopated words were used to avoid a loss of information.

In the second part of the process, a search was carried out in each of the databases using the most appropriate filters in each case to narrow down the research topic. Finally, a review of both titles and abstracts was carried out, making an initial selection based on the occurrence of the main descriptors “COVID”, “families”, or “education” accompanied by any of those mentioned above while considering the inclusion and exclusion criteria.

### 2.2. Inclusion and Exclusion Criteria

In this case, the inclusion criteria comprised works in which three of the main descriptors or variants were in the title, keywords, or abstract and works that could be accessed in the full text. This criterion, fortunately, proved to be possible in all the publications related to COVID-19, even in journals and publishers that do not usually publish in an open access format. The exclusion criteria included articles not written in English or Spanish, those written before 2019, those not related to family and home education caused by COVID-19, studies carried out on health without a link to education and the adaptation of families, articles unrelated to the subject of education, works that could not be accessed in full text, non-empirical studies, and those without further research that were more an explanation of a future project.

### 2.3. Screening and Selection Process

The screening and selection process was performed from February 2021 to May 2021 by two independent reviewers and supervised by a third reviewer to solve any possible discrepancy in study selection according to exclusion and inclusion criteria mentioned above. The final number of works used in the present study was 28, of which 80% (*n* = 20) were found in two or three databases. Only 20% (*n* = 5) were registered in a single database (2 in ERIC, 1 in WOS, 1 in Dialnet Plus, and 1 in Scopus). On the other hand, 85% (*n* = 21) of the reviewed papers were written in English, with Spanish being the language of the remaining 15% (*n* = 4). Figure 1 shows the search scheme for the various studies.

Table 1 presents a detailed description of each of the searches carried out, the Boolean operations used, filters, and the number of articles selected in each of the review stages.

## 3. Results

The main characteristics from the selected articles for the present systematic review are presented below. According to Table 2, most of the articles were focused on schools and families’ relationships during the COVID-19 pandemic, although some focused on psychological factors (well-being, stress, anxiety, etc.). In contrast, others analysed the educational response of families with children who have some type of disability.

Table 2 shows the total number of articles and the country where the research was conducted, the main research objective, the educational stage studied, and the relationships established.

One of the studies on family–school relations established the design and validation of an assessment instrument of said relationship [35], while the other studies applied a qualitative or quantitative methodology to obtain data.

Concerning the relationships between students and the school, only two articles focused in depth on these variables [28,50], though there were sample differences, since, the respondents in the first study were high school students and the respondents in the second study were infant and primary school pupils.

Some studies focused on the relationships between the three groups [12,20,51,52,53]. It is worth noting how the actions carried out in Norway are one of the best examples of collaborative relationships formed between families, teachers, and students.

All the studies used various technological resources such as the internet, Google Form, Facebook, and phone calls to administer online surveys and conduct in-depth interviews. This methodology was in line with health recommendations and advice but left out families with less or no access to electronic resources.

Additionally, some studies analysed variables related to the support that had been offered to students with disabilities during home-schooling; three of them mainly focused on the family–school relationship [54,55,56], another three mainly focused on the family–child relationship [24,57,58]; and a single study combined the three variables [59].

Several studies [11,18,37,60] addressed the relationships between the groups studied from a psychological perspective, emphasizing the participants’ stress, health, and well-being.

A lack of motivation or change in children’s behaviour are variables that have been studied in research focused on the family–child relationship [15,27,53,61].

### 3.1. Family–School Relationships during the Pandemic

For many families, the closure of schools meant converting their homes into classrooms. The most challenging aspects in seven of the analysed studies, in which families expressed feelings of frustration, concern, and denial, were: combining housework [11,57], the need to create or establish communication links with teachers to guarantee educational tutoring of their children [15,53,62], and managing and balancing the time spent on educational needs and that spent working either outside the home or working from home [27,51,63].

Regarding the decisions taken by national governments, the studies developed on Spain [20,28,62,64] showed that the various measures carried out have not satisfied the demands and necessities of either teachers or families and, on many occasions, they worked along different paths [28]. One of the articles carried out in Hong Kong stated that despite home learning being unanimously established [64], there were insufficient specific guidelines and schools had to take the lead. In contrast, another study centred on Kazakhstan [63] claimed that government granted schools the freedom to establish flexible approaches to facilitate learning for students, although the schools had to report on the success of their actions. One of the negative aspects of government efforts was excessive bureaucratization [59], where families with children with disabilities did not receive additional support or aid for carrying out therapies. Despite all these, governments decided to continue with their own established schedules or make minor modifications to the main ones [14,18,20,28,50,63]. Those changes were made in each country and, on several occasions, in each region or county separately and individually, without making a common decision at global scale but reorganizing education according to their own characteristics, which can explain the differences between studies.

**Table 2 ijerph-18-11710-t002:** List and characteristics of articles selected by the systematic review.

Relationship and Number		Authors	Country	Research Objective	Educational Stage
School–family relationship	1	Davis et al. (2020) [37]	USA	Parental anxiety	-
	2	Bokayev et al. (2021) [63]	Pan-Kazakhstan	Parental involvement, satisfaction, and quality of education	-
	3	Díez- Gutiérrez and Gajardo-Espinoza (2020) [20]	Spain	Perspective of families and students on education and assessment	-
	4	Hortigüela-Alcalá et al. (2020) [62]	Spain	Family–school, family–teacher, and family–student relationships Effects of virtual teaching	Infant primary, secondary, and university
	5	Yates et al. (2020) [59]	Australia	Investigate funds dedicated to people with disabilities and their development	-
	6	Pozas et al. (2021) [51]	Mexico and Germany	Home-schooling opportunities and challenges	Infant
	7	Jæger and Blaabæk (2020) [65]	Denmark	Inequalities in families regarding education	Infant, primary, secondary, and university
	8	Güvercin, Kesici and Akbaşlı (2021) [18]	Turkey	Changes, challenges, perceptions, and experiences of teachers and parents during pandemic	Infant, primary, secondary, and higher education
	9	Thorell et al. (2021) [54]	United States, Sweden, Spain, Belgium, Netherlands, Germany, Italy	Organization of home education; negative and positive experiences; comparison of families with children with some type of mental illness or difficulty	Nursery/preschool Primary and secondary
	10	Weaver and Swank (2020) [12]	America	Parental experiences	Infant primary, and secondary
	11	Lau and Lee (2020) [50]	Hong Kong	Parents’ opinion of distance learning Perceptions of difficulties and necessary support Children’s screen usage time	Infant primary
	12	Wendel et al. (2020) [55]	Canada	Changes in the child’s and parent’s behaviour	Infant
	13	Bonal and González (2020) [64]	Spain (Catalonia)	Learning gap between students of different social origins	Infant primary, and secondary
	14	Jones (2020) [13]	USA	Home learning, expectations, adjustments, challenges, and benefits, as well as the concerns of parents and teachers.	Infant primary
	15	Dong et al. (2020) [14]	China	Children’s experiences Parents’ beliefs about and attitudes towards learning	Infant primary
	16	Sosa (2021) [56]	Spain	Changes in education, socio–digital inequalities, and family participation and accompaniment	Infant, primary, secondary and special needs
School–family–student relationship	17	Cahapay (2020) [58]	Philippines	Opportunities, changes, and challenges for parents of children with autism	-
	18	Rojas (2020) [35]	Ecuador	Study of the parent–family relationship	Infant Primary, and secondary
	19	Sala (2020) [28]	Spain	Evaluate whether the students were able to follow the work remotely	High school
	20	Bubb and Jones (2020) [52]	Norway	Know the point of view of teachers, parents, and students about how teaching has developed during COVID-19	Infant primary, and secondary
	21	Yıldırım (2021) [53]	Turkey	Perceptions of teacher and parents about COVID-19 effect on preschool education, and changes in educative content.	Infant
Student–family relationship	22	Goldberg et al. (2020) [11]	USA	School–work relationships Stress and parental concerns about the pandemic	Infant, primary, and secondary
	23	Taubman-Ben-Ari and Ben-Yaakov (2020) [60]	Israel	Parental anxiety, stress, and apprehension	Infant
	24	Neece, McIntyre and Fenning (2020) [57]	USA (California and Oregon)	Parental perspectives of the impact on parents with young children with developmental delay or autism spectrum	-
	25	Majoko and Dudu (2020) [24]	Zimbabwe	Parent strategies for educating children with ADD Challenges and opportunities to home-schooled children	Primary and secondary
	26	Parczewska (2020) [15]	Lublin, Podlaskie Masovian and Greater Poland voivodeships	Parents’ experiences and difficulties	Infant primary
School family/ student–family relationship	27	Garbe et al. (2020) [27]	USA	Parents’ experiences and difficulties	Infant primary, secondary, and university
	28	Lee et al. (2021) [61]	USA	Analyse activities that parents carry out with their children, educational activities, and state of well-being	Infant primary

Schools have had to improvise based on their resources, thus generating a necessary two-way relationship with families to send the students’ homework. The level of involvement and the family–school relationship depend on the students’ educational stage [13]. Greater participation in online learning is observed in the compulsory, primary, and secondary educational stages [13,15] than in the non-compulsory stages, such as infant [14,50,53,55], professional and technical training courses, and university [20].

The establishment of communication channels between families and schools has been conditioned by the multitude of platforms and means of communication available such as Skype, Zoom, and WhatsApp, the latter being the most used application during the pandemic [18,35]. Spanish teachers [62] have stated that families do not know the virtual teaching model their children are using, although they did praise the communication channels established between teachers and families, which showed a considerable increase during the pandemic [35,56]. A similar situation occurred in a study carried out in Hong Kong [50], in which more than half of the parents were not satisfied with the support measures offered by the school but were satisfied with the actual learning activities proposed. On the other hand, many parents have reported the desire to communicate with teachers to receive guidance on how to proceed with the multitude of resources and online platforms provided [27,53].

In theory, the already consolidated family–school relationships should not be affected by the lockdown. However, the exceptional situation has catapulted one of the gaps that make it difficult for these relationships to flow. The digital divide [35,64] has meant that thousands of families have been unable to establish successful communications due to low levels of computer skills [14,56,64], difficultly in accessing the internet [53,61], and the time dedicated to using these digital means. A technological barrier to learning has been generated [27,56], though this can be eradicated by giving families greater access to technology to become more technologically proficient and thus help students carry out their tasks [63].

#### Family–School Relationships and Students with Disabilities

The pandemic seriously affected families and students with disabilities since it was quite complex to access educational resources, such as technological tools, the internet, and various devices (tablet or computer), at home [27,62]. This was acerbated by a lack of knowledge of the pedagogies carried out at school [27] or the ability to reproduce them at home, the lack of communication with teachers and specialists [27,57,59], a lack of support [24,57,59], and excessive bureaucratic obstruction to request the aid they had received before the pandemic [49].

Families with children with disabilities were largely overwhelmed, frustrated, and stressed [24,54,59] in the face of assuming new routines [55,57], as well as their children’s educational tasks at home, and not receiving any guidance on how they should manage the curriculum that was being carried out [54,56]. However, all family members were involved in the teaching–learning processes [24,54,55,56,57,58]. Parents went beyond serving as support to access the platforms where the didactic contents were housed [11,55] since on many occasions, the families saw their hours of homework support increased [59] and the need to adapt said content, since the teachers uploaded homogeneous materials without adaptations [24].

Despite all difficulties, some parents have found home-schooling to be a viable alternative to face-to-face education with multiple advantages for their children, since the relationships with the specialists were good throughout the lockdown and the students showed significant improvements in both knowledge and behaviour [24,54,58]. Furthermore, due to poor communication between teachers, many parents contacted other families in similar circumstances [24,56].

### 3.2. Teacher–Student Relationships in Times of Pandemic

The student–teacher relationship has gone from bidirectional in a face-to-face classroom environment to unidirectional in the online education sphere [62]. The relationships established in a pandemic were conditioned by short lessons [63] and the timely delivery of assigned tasks [64]. Many parents had to intensify their efforts [27] to deliver tasks of little importance or significance [62] because they were hosted on multiple and sometimes complex platforms available to teachers and students. A feedback system between students and teachers regarding the monitoring of work was established using alternative communication channels such as emails [63]. However, communication presented more negative aspects at higher educational levels such as secondary and university [20].

Concerning study plans and modifications of the curricula, teachers have had to make alterations to adapt face-to-face learning to virtual learning in order to [64] increase student participation [50,56], foster social relationships [51], offer support [27,53], provide feedback [13,28], and use the most appropriate educational platforms to improve the experience [65].

The situation caused by the COVID-19 pandemic has severely challenged teachers’ technological knowledge [15,18,63] regarding the development and creation of digital content, as well as the use of educational platforms adapted to different academic levels. They often chose to use pre-prepared or pre-recorded materials [14,50] hosted on different educational platforms, thus favouring an asynchronous education that facilitated the connection of students to this platform but sacrificed interaction in learning [14,51].

The issue of homework has been one of the turning points in this relationship [12]. Teachers have devoted more time, effort, and creativity [13] to carrying out assignments, class preparation, and question-solving [28,53] than to providing practical guides regarding the time required to do a task [27], since many students spent more than three hours a day in front of the screens to carry out homework, thus increasing the use of electronic devices [15,50,54,64]. Parents reported spending more than an hour a day supporting their children in order to continue with scheduled classes [14,27] and attributed this complex situation to excessive homework [18,20] and imposed requirements [15], feeling as if educational institutions were trying to recreate a school day without considering family consequences.

Another aspect to consider concerning homework is the difficulty of the tasks and the cognitive challenges they present. Research has emphasized the materials’ homogenization of tasks [51,59] or a lack of difficulty [14,27,56] compared to the level of tasks and activities set in face-to-face contexts.

There are conflicting opinions regarding the quality of education received by students during the pandemic. On the one hand, parents have stated that they have a favourable opinion about the quality of education [63,64] and the evaluations of home learning carried out during the pandemic [14]; the quality of teaching is prioritized over the way it is delivered [65]. On the other hand, many parents have expressed serious concerns regarding the quantity and quality of content provided to their children [12,18,27]; these worries have been aggravated when thinking about the possible resumption of face-to-face classes, about which a vast majority of parents expressed concern due to the low educational level their children had during home-schooling [11,12,20,50]. The link between schools and parents results from positive social behaviours and academic outcomes [55,64].

Two of the selected studies [51,52] stand out as resounding exceptions to the rest of the investigations. In the first of these studies [51], the home-based educational possibilities offered in Germany were very different from the rest of the analysed studies. The students received a differentiated instruction that adapted the educational activities and tasks to the characteristics and needs of each student, whether they had a disability or not. Parents experienced no difficulties when it came to accessing different electronic resources or maintaining a cordial relationship with the school. Moreover, the study also showed how home-schooling was not a barrier to develop inclusive approaches, as students received learning aids appropriate to their needs (extra time, extra homework, and daily plans). Research carried out in Norway [52] established a differentiating starting point from the rest of the research, stating that schools had laid the foundation for digital learning before home learning began so that all students had a tablet and were accustomed to its use in face-to-face settings. The teachers, for their part, had received the necessary training in their use and the municipality had invested the necessary capital in purchasing the resources, thus establishing an almost utopian link between school, students, and their social environment. Positive results were quickly shown due to the considerably increased use of new technologies; the holding of periodic meetings between families, teachers, and specialists; the possibility of offering feedback to their students; and the increase in digital competence of both teachers and students.

### 3.3. Family–Student Relationships during the Pandemic

Seven studies specifically addressed this family–child relationship since the sudden change in daily routines [11,57] and the responsibility assumed by families in taking up the teaching role within the home [24,48,64] have made dents in this relationship. Parents have reported feeling overwhelmed and distressed [12,18] by the situation, given their low qualification in this regard [27,37,51,56,63] and the need to continue working and carrying out household chores. As a consequence, situations of verbal violence [15,54,63], stress, and decreased general well-being have emerged in families [11,37,64], teachers [13,37], and students themselves [12,13,50,63].

Many families have reported that tasks are complicated when several children are in the home, each in different educational levels and with different needs [13,27,63]. A scarcity of resources [18,35,51,53], lack of time [12,13,27,63], and uncertainty surrounding the pandemic increased the stress of families in this situation [54].

Parents must establish communication channels with their children that allow them to set limits [50,61] on the independent use of technologies and encourage alternate activities (sports, video games, leisure, etc.) [15,18,61,64], whether or not it is specifically aimed at reducing screen time in favour of strengthening social ties and relationships [11] with different family members [50].

Student motivation has decreased as lockdown has lengthened over time due to the difficulty in using different educational platforms [12,13] or in living harmoniously with other family members [50] according to the analysed studies [27,53]. A lack of motivation, boredom, decreased attention span, concentration, or cooperation with their children have made it very difficult for parents to fulfil their responsibilities as they divide their attention between motivating one child’s learning while taking care of other children [27]. This situation is even more difficult if a child has a disability at home [59].

In a home learning situation, the provision of technological resources to meet the educational needs of children was one of the most important concerns that was reflected in 12 articles, since the vast majority of families claim to have access to very few resources [15,18,27,35,51,52,63,65] to carry out the various learning tasks, as well as feeling overwhelmed by the sheer number of tasks to be completed [27,51,52]. Students also require appropriate tools such as computers [24], tablets [28], the internet [20,53], mobile phones [12,56,64], and television [12,63] to access online education [15].

School administrations have been aware of these concerns and needs, and they have taken according measures. Hong Kong, for example, implemented an assistance program for the acquisition of electronic resources for low-income families [50,64]. In some regions of Spain [28], school institutions provided low-income families with tablets with internet access to enable students to attend virtual classes. In Zimbabwe [24], the ministry made digital services, learning platforms, and radios available to the population so that all students could receive a minimum level of education. Similarly, in Spain [20], educational television programs (5 h) were broadcast to increase resources and support student learning.

The relationships between families and students can be complex. Despite all the difficulties endured during the pandemic, however, ties with all family members have been strengthened [12,51,57,58]; many parents rediscovered their children, and children rediscovered their parents [13,15].

## 4. Discussion

This article has presented a systematic review of the most recent research on the relationships between students, families, and schools during COVID lockdown. The three groups were discussed in a comprehensive set of 25 articles.

In this context, we asked ourselves questions the following questions: (1) how supportive have the relationships between the family and the school and the school and the students been during the pandemic? (2) how have parents been involved in the task of educating their children during the pandemic? (3) have these three groups (family, school, and students) supported each other sufficiently? These questions are linked to the more general objective of presenting critical information about the relationship established between the family and the school in the face of an imposed distance education scenario due to COVID-19. We break it down into the following specific objectives: (1) to analyse what relationships have been established between the family and the school; (2) to determine how the school has supported its students; and (3) to examine what relationships have been constituted between parents and their children in a home learning situation.

### 4.1. Family–School Relationships during the Pandemic

Concerning the first objective, it was evident in a large number of investigations that one of the main concerns of families was related to time compatibility [27,51,63] to meet the educational needs of students [51,54,57,59] and be able to go to work or work from home. Similar studies showed that families sometimes have not had the sufficient capacity to be able to combine everything [56,66], coupled with the need to share resources [20,27,62] and spaces [15].

School administrations have acquired a special prominence within the development of home learning. It has been observed that the decisions made in some countries have not been those expected or desired [28,50,59,62]. The studies analysed in this review and others developed during the pandemic highlight the lack of coordination among school administrations in carrying out their responsibilities [67]. In addition, different approaches carried out by the administrations have made it difficult for schools to provide quality education, as shown by the non-governmental organization Save the Children [68].

One of the main lines of action of school administrations resides in reducing inequalities and allowing students with disabilities to have equal access to education and diverse activities by providing them with the necessary support. The present situation has revealed many shortcomings, as shown in another study carried out during the pandemic [69]. Many families have had to buy the necessary resources to continue their children’s therapies without government aid or benefits [59].

Both the good and the bad have been exposed in the relationships established between teachers and families. Depending on the educational stage and the country, these relationships have been either positive [27,52,62] or negative [14,19,20,24,51,54,55,57,64]. Indeed, they have mostly leaned toward the negative and exposing the need for improvement. Similar studies revealed that a vast majority of parents were not entirely satisfied with the relationships established between school and families [70], with regards both to online learning [71] and the difficulties in participating in it [72]. This dissatisfaction may be affected by the parent’s educational level, socioeconomic characteristics [73], or degree of involvement [74].

Regarding the offered resources, families have demonstrated little or no knowledge about the use of different technological tools and about virtual teaching [14,62,64], which shows that despite the dizzying pace with which they have adapted to these technologies, they have been less focused on teaching or pedagogical purposes [75] and more geared towards entertainment and leisure [62]. Furthermore, parents felt overwhelmed by the number of technological resources presented by the school [27,51,52]. Studies carried out before the pandemic already highlighted some of the challenges in this regard such as the establishment of communication channels between families and schools [76], the importance of teachers [77], financial resources [78], the lack of interest in the use of technology [79], the high level of commitment of families [80], and the establishment of a good two-way relationship between these two educational agents, all of which can positively influence both motivation and student academic performance [81,82,83]. Resources are an essential part of the family–school–student relationship, as demonstrated in research carried out during the pandemic [84].

It is quite clear that establishing communication channels between families and schools is essential if the aim is to successfully move towards learning, including online learning. Even before the pandemic, studies have shown the many positive aspects of virtual learning, such as direct interaction with teachers, which is more flexible than face-to-face learning [85]. Various studies have shown that online education should not be based solely on uploading and downloading documents or videos from different virtual platforms [85], nor should it be based on training and innovations; instead, it is crucial to train families in digital skills. This should be conducted is in addition to training teachers in the use of these resources [35,64] so that they can create scenarios appropriate to the needs and characteristics of their students and promote the different didactic strategies so that students achieve the desired meaningful learning—autonomous learning adapted to the rhythm of each one of them [86,87].

Special mention must be made of students with disabilities, since several of the analysed studies [27,51,59,62] showed that, in a pandemic situation and compared to fellow pupils, these groups are at a disadvantage in terms of education and well-being. This is not a new finding since, in previous studies, this group’s educational, social, and employment differences have been highlighted [88].

Some parents [24,54,57,58] noticed substantial improvements in the development and learning of their children with disabilities. Many ultimately decided to implement home-schooling as a definite rather than temporary measure. Research carried out during the pandemic showed that many families, despite its many challenges, have opted for home-schooling for their children [89].

As families have become “teachers” in the home learning environment, the critical role that teachers play in their children’s education, as well as the lack of preparation of parents and families to assume this role, has become increasingly evident [15,27]. Many parents have felt overwhelmed because their attention was split between their other responsibilities and having to master new technologies. Moreover, they fought with other issues exacerbated by the pandemic, such as anxiety, frustration, anger, irritation, fear, uncertainty, confusion, and loneliness [11,15,37,60]. In studies carried out during lockdown, it was shown that the role of the teacher is essential for supporting students, clarifying concepts, and deepening their understanding, thus releasing parents from this burden—actions which have been diminished during this period [56,66]. It is also important to remember that although parents are one of the main axes in the educational processes of their children, they do not have the necessary skills to promote knowledge acquisition [15,56].

### 4.2. Teacher–Student Relationships in Times of Pandemic

Concerning the second objective, student–teacher relationships have been overshadowed at all education levels and contexts by the exceptional situation that education has undergone. Students, and failing that, parents, must fully understand the management of multiple platforms in which teachers host content. In many cases, these platforms offer a few cognitive challenges for students [14,27], who must then wait until the next virtual class to contact the teacher and communicate any doubts they may have had [63], turning the teaching–learning process into a tedious and unmotivating activity [27]. Studies before the pandemic showed that the variety of activities and the feedback that students receive from teachers throughout the teaching–learning processes are vital for promoting meaningful learning [36,90].

It has been shown how the student–teacher relationship in compulsory schooling stages, whether primary or secondary [13,15], has become more robust in that teachers provide necessary wake-up calls when there have been connectivity problems or decreases in task productivity [27]. However, in the non-compulsory stages such as infant school or university levels [20,50,55], this communication has sometimes faltered [20]. However, in studies carried out before the pandemic, it was observed that families present a multitude of issues that entangles these communications (such as materials, resources, low culture, and different languages) [87].

One issue, present in most of the analysed studies and which parents have highlighted as being a major concern, is the constant use of computers [24], tablets [28], or mobile devices [12], as well as the amount of time spent doing school tasks. Pre-pandemic research analysed how excessive screen time use can negatively influence student development, increase health problems [14], and increase risk of accessing inappropriate content [82]. Some articles [50] have shown that the time spent by students carrying out their tasks exceeds 2–3 h a day, an aspect that goes against the recommendations of the World Health Organization (WHO) that argues that screen-time for children under five years of age should not be greater than one hour per day [91]. Although electronic or virtual learning has long been promoted [92,93], the pandemic has forced its implementation for an extended period and in such a generalized way for the first time.

### 4.3. Family–Student Relationships during the Pandemic

Finally, regarding the third objective, the changing relationship between parents and children has meant that parents’ levels of stress and frustration increased, as did feeling overwhelmed by the ever more difficult school tasks with which children needed help [10]. Thus, in research carried out before the pandemic, it was evident that the involvement of parents when tackling their children’s learning difficulties was also conditioned by face-to-face education, a condition that became harder to tackle when the learning took place in an online environment [89].

Some studies have shown how family violence or verbal violence between the family and students has increased during this lockdown [15,54,63], a regrettable fact but not an isolated issue since another study carried out during this period corroborated it [94].

**Limitations and suggestions**: Generally speaking, most of the selected studies were characterized by the fact that the samples were collected and accessed through social networks or the use of technologies. Thus, it is possible to perceive a bias in most of them since they excluded the most vulnerable families and those who do not have full access to the internet or computer resources due to low socioeconomic resources. On the other hand, studies that were written in languages other than English or Spanish were not selected for the present review, which may have been a source of selection bias as the epicentre of the pandemic was developed in China, a country were the mother language is not considered in school–family–students relationship as a trinomial. An additional limitation was the general lack of research directly centred on the opinion of students, especially of those at the highest educative levels. Understanding their experience is as important as the perceptions of families and teachers in order to know the real impact of the pandemic on the educative process. Although PRISMA guidelines were followed to complete every section and report data, it should be considered that analysed studies had both quantitative and qualitative methodologies. This fact could be perceived as both a strength and a limitation. Though the variety of methodology allowed for the analysis of the changes and relations between schools–families–students in a greater breadth, outcomes reflected on the present review were obtained through the general comparison of the different results of selected papers without working on the meta-analysis. Thus, the results of this review could be employed as the basis for future meta-analysis and empirical research while considering the limitations discussed above. It is necessary to keep studying the characteristics of educative systems of different countries, studying the measures of different government and educative centres, and analysing the differences that may be present between them. The present study proves the great variety of opinions, experiences, and perceptions around teaching–learning process progress in a limit situation. Thus, knowing the strengths and weaknesses of each country could help to improve educative systems around the world.

## 5. Conclusions

The coronavirus pandemic abruptly and suddenly changed the routines and prospects for many households around the world. The educational field was one of the most affected in this sense since after the successive closure of schools worldwide in mid-March, an alternative plan to the acclaimed and entrenched face-to-face education needed to be improvised.

This systematic review revealed an objective reality: in the 21st century, students’ lack of autonomy and motivation is attached to an educational system that continually revolves around face-to-face education.

New technologies have been the immediate and most effective solution to the closure of schools, thus becoming both a problem and solution regarding a complex social and educational situation. They have evidenced various inconsistencies and setbacks that had remained hidden under the normality of pre-academic education, such as the enormous challenge posed by its immediate use within a purely face-to-face educational system, the scant training of families in its use, the limited access to it by many students, and the diversity of platforms and media.

Schools and families have had to strengthen their relationships, fight for their causes, and satisfy their students’ educational needs. Parents and their children have discovered various positive and negative effects of home-schooling, though the adverse effects have been much more palpable and evident. A beneficial line of future research may be related to those positive aspects of home-schooling that need further study.

While families and students have had time to experience the effects of home-schooling, educational institutions have had the opportunity to rethink how education is delivered. They must seriously consider both the challenges and the opportunities that online education brings without leaving behind the different groups that, due to their characteristics (such as low socioeconomic level, disability, or ethnic minorities), are more vulnerable and, unfortunately, have been forgotten in pandemic education.

## Figures and Tables

**Figure 1 ijerph-18-11710-f001:**
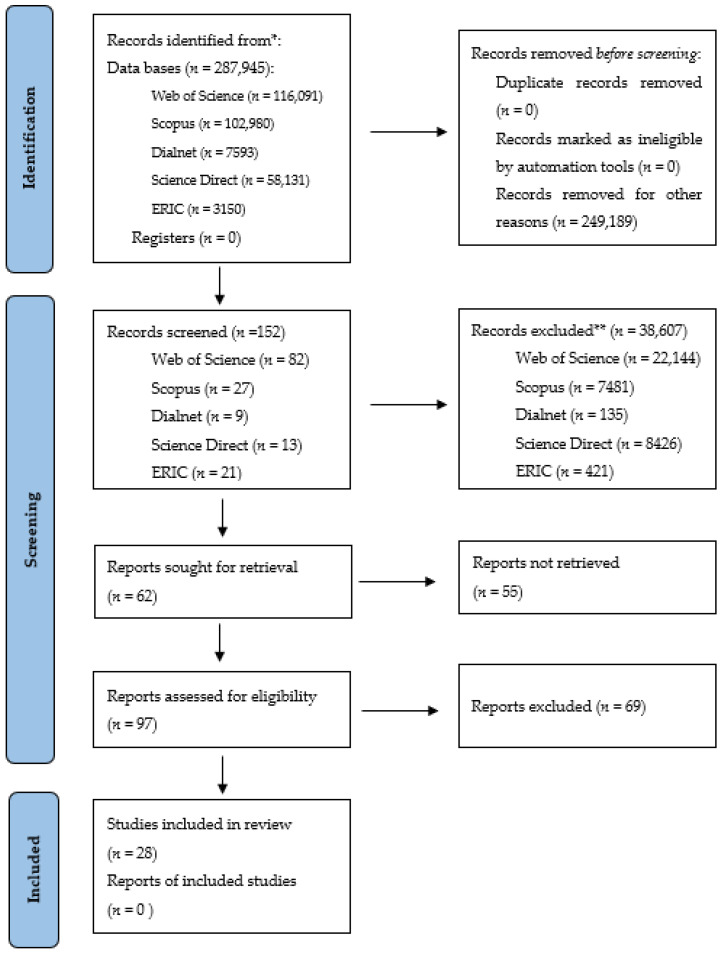
Selection criteria flow chart 1.

**Table 1 ijerph-18-11710-t001:** Procedure for selecting articles from the structured search in the primary databases.

Database	Boolean Operations	Initial Number	Filters	After Filters	After Criteria	Final
**WoS**	COVID-19	105,585	Domain: social sciences; databases: Web of Science core collection; languages: English and Spanish; research areas: education, educational research, social issues, family studies, social work, and sociology.	19,453	79	22
	COVID-19 AND famil *	2945		1523		
	COVID-19 AND educa * AND famil *	411		301		
	COVID-19 AND (paren * or pad *) AND educa *	149		42		
	COVID-19 AND educa * AND (youth OR child * Or adolesc * OR young OR niñ *)	514		95		
	COVID-19 AND school *	1451		442		
	COVID-19 AND school * AND famil *	310		122		
	COVID-19 AND school * AND (youth OR child * Or adolesc * OR young OR niñ *)	730		227		
	COVID-19 and home-schooling	18	No filters; all checked.	18		
**SCOPUS**	COVID-19	92,282	Search within: article title, abstract, keywords; years: 2020 and 2021; subject area: social sciences and psychology (exclude the rest); language: English and Spanish.	4473	27	21
	COVID-19 AND famil *	4280		957		
	COVID-19 AND educa * AND famil *	692		233		
	COVID-19 AND (paren * or pad *) AND educa *	258		122		
	COVID-19 AND educa * AND (youth OR child * Or adolesc * OR young OR niñ *)	1332		390		
	COVID-19 AND school *	2638		890		
	COVID-19 AND school * AND famil *	376		138		
	COVID-19 AND school * AND (youth OR child * Or adolesc * OR young OR niñ *)	1098		291		
	COVID-19 and home-schooling	24		14		
**Dialnet plus**	COVID-19	6210	Filters: social sciences, psychology, and education; languages: Spanish and English	80	9	3
	COVID-19 AND famil *	265		5		
	COVID-19 AND educa * AND famil *	73		5		
	COVID-19 AND (paren * or pad *) AND educa *	817		35		
	COVID-19 AND educa * AND (adolescent* OR child* Or adolesc * OR niñ *)	38		4		
	COVID-19 AND school * or escuel *	158		10		
	COVID-19 AND school * AND famil *	28		3		
	COVID-19 AND school * AND (youth OR child * Or adolesc * OR young OR niñ *)	3		1		
	COVID-19 and home-schooling	1		1		
**Science Direct**	COVID-19	40,220	Subject areas: social sciences and psychology	4521	13	3
	COVID-19 AND family	11,553		2013		
	COVID-19 AND education AND family or familia	4390		1172		
	COVID-19 AND parent AND education	1441		546		
	COVID-19 AND education AND adolescent OR child	468		169		
	COVID-19 AND school OR escuela	31		7		
	COVID-19 AND education AND family or familia	16		7		
	COVID-19 and home-schooling	12		4		
**RIC**	COVID-19	1476	Audience: parents	11	21	4
	COVID-19 AND family	227	Descriptor: distance education	100		
	COVID-19 AND education AND family or familia	205		100		
	COVID-19 AND parent AND education	148		80		
	COVID-19 AND education AND adolescent OR child	17		6		
	COVID-19 AND school OR escuela	886		42		
	COVID-19 AND education AND family or familia	182		94		
	COVID-19 and home-schooling	9		9		

* appocoped words.

## Data Availability

Not applicable.

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
