# Peer review of "Family and School Relationship during COVID-19 Pandemic: A Systematic Review"

_ijerph, 2021, doi:10.3390/ijerph182111710_

Round 1

Reviewer 1 Report

This article addresses a very relevant topic in the current situation due to the impact of Covid-19 among schools and families. It is very important to know how schools and families have reacted towards the challenges posed by the pandemia. However, there are some important inconsistencies in your paper:

(1) Lines 37-38. You claim that students at all educational levels were forced to turn their homes. Is this the situation for everyone? In my knowledge, not all the countries (even regions within the same country) adopted the same measures. You may need to adjust your claim to the diversity of situations. The same applies for the words "purely virtual". There are many differences between countries/regions. 

(2) Line 91. Among the words in English, you write "familiar", which is Spanish. The word "familiar" in English does not have anything to do with "family" or "familiar" (relativo a "familia", but different meaning in English). 

(3) I suggest to do language editing of the document (i.e., line 113 "was 25", must to be "were 25".

(4) Figure 1 (line 121, etc.) is not clear at all. For instance: if you go from "Records screened (n=149)" in "screening" to Reports sought for retrieval (n=62) and Reports not retrieved (n=55), the addition is not equal to 149 (62-55=117). Where is the difference (32)?

Then, you go again to 149 in the next "box" (reports assessed for eligibility), and you add a new box with another 124 cases... which makes even harder to understand the diagram. 

(5) Line 129. There is no heading. Are we now in section "Results"? "Findings"? It is very difficult for the reader to follow the flow of your paper. 

(6) You claim that relationships between families and schools are a topic of interest in almost al continents, but Antarctica. It is really surprising that you are considering Antarctica... How much people live in Antarctica? I really wonder if there are regular schools and families sending their children to the schools, over there... 

(7) Line 150. You claim: This data is striking because the pandemic's episode epicentre is located in China. However, your claim is very surprising since you reported that articles other than in Spanish or English were excluded from the sample. Then, it seems quite obvious that the main number of articles are from America and Europe. 

(8) Lines 172... You talk about the situation in different countries (Hong Kong, Kazahstan, Australia, etc.) making general claims but what is the basis for that? Have you conducted an statistical representative study in each country (also making sure that sample is selected randomly) to be able to make such general claims? I would recommend to stay on the scope of the paper that you are quoting, not assuming that that findings can be generalized to a whole country. 

Summarizing, I think that this article needs major revision, because there are some assumptions that we cannot accept in a scientific paper as this is (for instance, the generalization, or the attribution of causality to findings that are motivated for the way in which the data collection procedure has been conducted, for instance). After checking all of this, I think that the authors must resubmit their paper for additional review, because the topic is highly relevant for the audience. 

Author Response

Dear Reviewer

We appreciate all your indications for the improvement of the article. They have helped us a lot to increase its quality.
Below we detail all the modifications made and their explanation.

  1. Lines 37-38 à Information provided has been changed in order to meet the reality according to the different measures adopted by countries worldwide to ensure educative progress during the pandemic. We had into account the possibility of developing presential classes, blended learning, and entirely virtual learning during this period.

  1. The Word “familiar” has been deleted from the English descriptors as it does not modify the results. It could be found the same amount of document with the world “family” or “famil*”, both used in the main search.

  1. A language editing has been done and English language of the whole document has been revised

  1. Thanks to open access, it was possible to obtain some of the documents directly on the first review. We considered that only the documents that had to be requested from the authors to obtain access to the full text should be considered in this section. Thus, out of 149 articles, 87 were directly accessible thanks to open access and 62 had to be requested from the authors, 55 of which could not be accessed. In the case of a misinterpretation, the number of articles would be changed to 149, of which 55 were eliminated.

The second part of the Flow diagram “reports assessed for eligibility”, 149 articles were assessed for eligibility. We could not find 55 of them and 69 remained did not meet the criteria (69+55=124 excluded). However, if there is any misunderstanding of the headings of the figure, data can be changed.

  1. Paragraph from line 121 to 123 from section “Screening and selection process” specifies that “table 1 presents the detailed description of each of the searches carried out, the Boolean operations used, filters, and the number of articles selected in each of the review stages”. Table 1 has the heading “Table 1. Procedure for selecting articles from the structured search in the primary databases” Thus, table 1 corresponds to “Screening and selection process” sections and it completes the Flow diagram shown before it.

  1. As we are not revising continent but publications and its origin, Antarctica has been included in the analysis to avoid bias and its exclusion even though it is a continent. It is a fact that we did not find any publication about covid-19, families and education conducted in the Antarctica even though it can be found other kind of research about covid-19 in this continent. We have considered all the possibilities

  1. Paragraph from 151 to 158 has been eliminated as the results presented on it only refers to articles on English and Spanish.

  1. The results shown in the present review is based on the results and the information provided by the articles and studies selected, which have carried out statistical analysis on each of the countries mentioned in the methodology. However, the information provided in the results of the review has been changed to mention the scope of the main studies as concrete results but no as a general claim for the country.

Reviewer 2 Report

This manuscript presents an interesting research, based on a meta-analysis of scientific articles of contrasted quality, focused on the relationship between the family and the school in the scenario of the new didactic dynamics required by the COVID-19 epidemic.

The introduction is brief, but it includes the problem according to the scientific literature analyzed. The methodology and results sections are well presented, with rigor and quality, including illustrative graphs that are effective for the reader. Finally, the discussion and conclusion sections also respond adequately to the objectives of this research.

In general, we would recommend the following modifications and additions:

  1. Revision of the references. Some do not meet MDPI standards.
  2. Since this is a topical subject centered on the COVID-19 epidemic, we would advise to consult more articles published in 2021 and related to the studied problem. The reason is that only 10 of the 81 references refer to this year. A few more from this year could be included.

We consider it an article of great interest for the journal, and we would recommend its publication. We thank you for the opportunity to have been able to carry out this review.

Author Response

Dear Reviewer

We appreciate all your indications for the improvement of the article. They have helped us a lot to increase its quality.
Below we detail all the modifications made and their explanation.

  1. The references have been reviewed so they can meet MDPI standards
  2. There have been included three more articles in the revision of the literature that have been published in 2021

Reviewer 3 Report

Dear Authors,

thank you for sending your manuscripts titled "Family and school relationship during COVID-19 pandemic: a systematic review"

The topic is extremely interesting and the review is necessary to look at the research gap from a broader perspective.

Some modifications are necessary in order to progress

  • Introduction
    • The introduction is extremely brief and should cover more aspects related to the pandemic, the impact that it had on children's, families and tutors/teachers. It suggested structuring the introduction with the following sub-section: 1.1 Family perspective, 1.2 Students perspective, 1.3 Teachers perspective. In this way, you will immediately help the reader understanding the 3 key areas of interest in your review and by citing previous work you can build the foundation for your review.
  • Methods
    • You methods the PRISMA strategies. However, there is no reference in the bibliography.
    • You must include and report that you have followed the PRISMA guidelines. Please read: http://www.prisma-statement.org/documents/PRISMA_2020_checklist.pdf
  • Results
    • The results appear well presented. However, the table needs to be reorganised so that they can fit on one page and became easier for the readers to interpret.
    • If you structure the introduction as suggested you will be able to maintain the same structure in the results looking at Teachers-Families-Students perspectives
    • Please follow PRISMA guidelines in how to report data
  • Discussion
    • Please follow the same structure of the introduction. Define clearly the aim of your manuscript and then the research question
    • Include a "take-home or Key points" table in your discussion
    • Include more details about future research and suggestion to facilitate learning outcomes 
    • Provide strategies for the current semester of blended approach (put your review in perspective with today scenario).
  • Limitations
    • More details are necessary about the limitation and what would you do differently if you are running the same study again

Thank you best wishes

Author Response

Dear Reviewer

We appreciate all your indications for the improvement of the article. They have helped us a lot to increase its quality.
Below we detail all the modifications made and their explanation.

  1. We have separated the introduction according to the subsection that were specified in the result section. We have included more information on it, taking into account some new articles that refers to 2021.
  2. PRISMA reference have been included within the de bibliography following MDPI standards
  3. We have followed PRISMA guidelines to do the present systematic review and we have reported it in the methodology, alluding to its reference.
  4. We have tried to reorganize the content of the tables so they can fit in one page, but it was not possible to do so due to the large quantity of information that is reflected on the tables and the necessity of following a concrete format. However, we have grouped the articles so it can take less space.
  5. We have changed the information so the structure of the introduction, the results and the discussion could match.
  6. We have followed PRISMA guidelines to report data. We have re-verified Prisma guideline to make sure all sections comply with the correspondent item taking into account we did not make a meta-analysis study.
  7. We have included a table in the discussion that identifies the key point of the review
  8. We have included some suggestions to future research in the section of “limitation and suggestion”, as well as for facilitating learning outcomes and strategies for a blended approach.
  9. Limitation: we have included more details for the limitation as well as some information about things we would change for future reviews

Reviewer 4 Report

Dear authors,

your study is valuable but I have some comments that you might address:

1) You review paper is short in analysing the most significant papers in the area. The use of three descriptors in the searching analysis have resulted in a shortage or strong discrimination of biliography. In my area of research, a  review paper is provided for 150+ papers. In view of that restriction, going to a different analysis may result in a big problem. Therefore, coud you add more papers (+10 or those known by you)  that you may know, I mean, from your expertise, to provide a more critical analysis?

2) To me, Table 1 should appear in an Appendix section but not on the main body of the manuscript.

3) Table 2 is too much lineal. I would prefer to present a table with the 'Established relationships' on the first column, with grouping them, and the list of references that are connected with each category.

4) The categories in the results section are very valuable.

5) The discussion is sound. The subsection 'Limitations' but is not offering a piece of criticism. Please, add sound limitations.

Author Response

Dear Reviewer

We appreciate all your indications for the improvement of the article. They have helped us a lot to increase its quality.
Below we detail all the modifications made and their explanation.

  • We have included three more studies within the results making a secondary search following PRISMA statements. We could not find any other study that follows inclusion and exclusion criteria to meet PRISMA guidelines, so that the analysis maintains the robustness and its scientific value.
  • We have maintained Table 1 in methodology sections as it complements the flow diagram above. We think that it could be easier for the reader to follow the review structure if we keep the table following the PRISMA chart.
  • We have change Table 2 distribution so all studies within the same category of relationships are all together.
  • we have included more details for the limitation as well as some information about things we would change for future reviews

Round 2

Reviewer 1 Report

I think that now the article looks better. Previous lacks of information, gaps, etc., have been addressed in this new version. I think that as it is now, it could contribute to the field. 

Reviewer 3 Report

Dear Authors,

thank you for coming back with the correct modification to your manuscript. I believe it is now suitable for publication.

Best wishes.

Reviewer 4 Report

Dear authors,

the comments have been precesily answered. Therefore, I recommend publication.